# Elite VABB 13G: A New Ultrasound-Guided Wireless Biopsy System for Breast Lesions. Technical Characteristics and Comparison with Respect to Traditional Core-Biopsy 14–16G Systems

**DOI:** 10.3390/diagnostics10050291

**Published:** 2020-05-09

**Authors:** Daniele La Forgia, Alfonso Fausto, Gianluca Gatta, Graziella Di Grezia, Angela Faggian, Annarita Fanizzi, Daniela Cutrignelli, Rosalba Dentamaro, Vittorio Didonna, Vito Lorusso, Raffaella Massafra, Sabina Tangaro, Maria Antonietta Mazzei

**Affiliations:** 1Radiodiagnostica Senologica, I.R.C.C.S. Istituto Tumori “Giovanni Paolo II”, Viale Orazio Flacco 65, 70124 Bari, Italy; d.laforgia@oncologico.bari.it (D.L.F.); rosalbadentamaro@libero.it (R.D.); 2Dipartimento di Diagnostica per Immagini, Azienda Ospedaliera Universitaria Senese, Viale Bracci 10, 53100 Siena, Italy; afausto@sirm.org (A.F.); alfofa@yahoo.it (M.A.M.); 3Dipartimento Medicina di Precisione, Università degli Studi della Campania Luigi Vanvitelli, Piazza L. Miraglia 2, 80138 Napoli, Italy; ggatta@sirm.org; 4Dipartimento dei Servizi—Diagnostica per Immagini, Ospedale “G. Criscuoli”, Via Quadrivio, 83054 Avellino, Italy; graziella.digrezia@gmail.com; 5UOC Diagnostica per Immagini, Azienda Ospedaliera San Pio, Via dell’Angelo 1, 82100 Benevento, Italy; angela.faggian@libero.it; 6Oncologia Medica, I.R.C.C.S. Istituto Tumori “Giovanni Paolo II”, Viale Orazio Flacco 65, 70124 Bari, Italy; vitolorusso@me.com; 7Chirurgia Plastica, I.R.C.C.S. Istituto Tumori “Giovanni Paolo II”, Viale Orazio Flacco 65, 70124 Bari, Italy; daniela.cutrignelli@gmail.com; 8Fisica Medica, I.R.C.C.S. Istituto Tumori “Giovanni Paolo II”, Viale Orazio Flacco 65, 70124 Bari, Italy; v.didonna@oncologico.bari.it (V.D.); massafraraffaella@gmail.com (R.M.); 9Dipartimento di Scienze del Suolo, della Pianta e degli Alimenti, Università degli Studi di Bari ‘Aldo Moro’, 70125 Bari, Italy; sonia.tangaro@ba.infn.it; 10Istituto Nazionale di Fisica Nucleare, Sezione di Bari, Via Giovanni Amendola, 165/a, 70126 Bari, Italy

**Keywords:** breast neoplasms, large-core needle biopsy, fine-needle biopsy, cytological techniques, endoscopic ultrasound-guided fine-needle aspiration

## Abstract

The typification of breast lumps with fine-needle biopsies is often affected by inconclusive results that extend diagnostic time. Many breast centers have progressively substituted cytology with micro-histology. The aim of this study is to assess the performance of a 13G-needle biopsy using cable-free vacuum-assisted breast biopsy (VABB) technology. Two of our operators carried out 200 micro-histological biopsies using the Elite 13G-needle VABB and 1314 14–16G-needle core biopsies (CBs) on BI-RADS 3, 4, and 5 lesions. Thirty-one of the procedures were repeated following CB, eighteen following cytological biopsy, and three after undergoing both procedures. The VABB Elite procedure showed high diagnostic performance with an accuracy of 94.00%, a sensitivity of 92.30%, and a specificity of 100%, while the diagnostic underestimation was 11.00%, all significantly comparable to of the CB procedure. The VABB Elite 13G system has been shown to be a simple, rapid, reliable, and well-tolerated biopsy procedure, without any significant complications and with a diagnostic performance comparable to traditional CB procedures. The histological class change in an extremely high number of samples would suggest the use of this procedure as a second-line biopsy for suspect cases or those with indeterminate cyto-histological results.

## 1. Introduction

Cytological and histological needle biopsies represent a fundamental step in breast diagnosis. Since the early 1990s, fine-needle aspiration cytology (FNAC) has become the most widespread method thanks to its simplicity, rapid execution, low cost, high sensitivity, and it being the best choice in anticoagulated patients [1,2,3]. However, over the years, this procedure has also shown some limitations connected with a number of false negatives (FNs), inadequate biopsies, operator-sensitive results, limited information regarding the tumor histology [4,5,6,7,8,9,10,11,12,13] above all in large lesions, in the diagnosis of microcalcifications, differentiation between invasive and in situ cancer, as well as between benign and malignant papillary lesions. The large number of FNs, the inadequate samples, operator-sensitive results, and the limited information obtained on the histology of the cancer using cytology [1,2,3,4,5,6,7,8,9,10,11] have, over the years, brought about a progressive substitution of FNAC with micro-histological sampling. Brancato [12], in line with other publications, identified a significantly higher rate of inadequate results for FNAC (17.07% vs. 6.10%) but the values for sensitivity, high specificity, and diagnostic accuracy almost overlap in the FNAC and core biopsy (CB) procedures, [8,9,10,11,12]. To date, the evidence present in literature continues to recommend FNAC as the most sensitive screening for breast cancer metastases in the axillary region [14,15,16].

Some studies highlight the reduced diagnostic accuracy of stereotactic guided fine-needle sampling of breast lesions compared to micro-histology using core biopsy (CB) or vacuum-assisted breast biopsy (VABB) [3,12,13].

According to the literature on micro-histological sampling, the VABB systems offer a lower underestimation compared to the CB [17,18,19,20,21,22,23,24], a better diagnostic accuracy, a higher negative predictive value (NPV), and an opportunity for the total excision of certain lesions [17,25,26]. These results are related to both the characteristics of the sampling and the needle gauge, which is generally larger in VABB biopsies. In literature, an instrument to assess the effectiveness and accuracy of a biopsy system is that of the underestimation of lesions considered at risk, mainly B3, such as atypical ductal hyperplasia (ADH) [14].

Brennan [18] in a meta-analysis of 52 studies and 7350 patients with ductal cancer in situ (DCIS) reported a 30.03% underestimation of the invasive ductal carcinoma (IDC) performed using a 14G-needle compared to 18.09% for the 11G-needle VABB procedure.

Several authors [19,20,21,22,23,24] report different percentages of underestimation between the CB and the ultrasound-guided VABB, which are significantly higher for the CB.

Moreover, according to the most recent publications of consensus [27,28,29] an important role emerges in the second-line VABB procedure using a 7–8G needle instead of excisional biopsy surgery: this would allow the complete removal of small B3 lesions and the exclusion of invasive carcinomas in several lesions that have already undergone core biopsy.

From the abovementioned, the need emerges to select biopsy systems with an ever-better ratio as regards performance, costs, and manageability, especially in ultrasound.

The available studies in literature, however, compare VABB 7–11G and CB systems or stereotaxic VABB systems among each other: though, as far as we know, there is no comparison between ultrasound-guided CB and low-gauge VABB systems on breast lesions.

To date, the available VABB devices with stereotactic, ultrasound- and MRI-guided systems perform better than CB, but are generally far bulkier and unwieldy, especially in ultrasound: this constitutes a major limitation in their use with this type of system, which is, moreover, the simplest and most easily available. The VABB cable-free systems could represent a good ratio between performance and manageability: indeed a new generation of 13G VABB needles (Mammotome Elite, Leica Biosystems, Newcastle, UK) has been introduced for ultrasound using a slightly larger handpiece than the one used in a CB.

The aim of the present study is to compare the results of a 13G-needle VABB Elite to traditional 14–16G CB systems and to the gold standard of conclusive histological diagnosis in order to evaluate its accuracy (primary endpoint).

The secondary endpoint is to assess the percentage of inadequate and inconclusive results and the diagnostic underestimation of both procedures.

## 2. Materials and Methods

### 2.1. Experimental Data

From January 2016 to October 2019, two breast radiologists, with over 15 years of experience, carried out, comprehensively, 1314 consecutive 14–16G-needle CBs and 200 13G-needle Elite VABBs (Mammotome Elite, Leica Biosystems, Newcastle, UK) using ultrasound.

In all, 1407 symptomatic or asymptomatic women (1211 for CB and 196 for Elite) aged between 25–83, were recruited for the study. All patients showed a visible breast anomaly detected by ultrasound and classified as BI-RADS 3, 4, and 5 lesions, and their written consent for the biopsy was obtained. The patients aged 40 or over also underwent a mammography. Patients with no visible ultrasound findings and those who had refused to give their consent were excluded.

Of the 196 patients examined with VABB Elite, 31 also underwent 14–16G semi-automatic core needle biopsy, 18 underwent FNAC cytological assessment, and 3 underwent both procedures. The radiological data were compared with the cytological FNAC and the micro-histological CB and VABB Elite biopsy results and then classified from 1 to 5 according to the common classifications in literature: class 1 for normal tissue (or inadequate biopsy), class 2 for benign lesions, class 3 for suspect lesions, class 4 for suspected malignancy, and class 5 for malignant lesions [30,31,32].

In the presence of ultrasound anomaly, the findings were typified using CB as a first-line investigation, where possible; whereas, VABB Elite biopsy was carried out in the following circumstances:a.Difficult breast site for core-biopsy (pre-pectoral or retroareolar regions)b.Indeterminate first cyto-histological result (classes 1 and 3)c.For total excision (for lesions up to 5 mm)d.The need for bigger biopsies in non-mass breast lesions with undefined marginse.Second-line biopsy in cases of radio-pathology discrepancy.

Both CB and VABB procedures are available on the market and are used in normal daily clinical practice.

The suspension of the administration of anticoagulant and antiplatelet therapy is required 3 days before the procedure, and all patients were administered a subcutaneous injection of 5–10 mL of lidocaine pre-biopsy. All patients signed a consent form for the procedure and for the authorization of data dissemination for scientific purposes. The procedure was also authorized by the scientific board of the institute.

### 2.2. VABB Elite Procedures

The ultrasound VABB Elite biopsy was performed using a sterile disposable 13G 2.6 mm probe, 13.6 cm in length and with a biopsy site 18.4 mm in width which, thanks to a cable-free and more compact handpiece, compares favorably with the simplicity and functionality of the CB (Figure 1). In Figure 2, we summarize the workflow of the VABB Elite procedure. The straightforward insertion of the needle into the breast is carried out only once, by means of a flat blade end and a double-lumen probe connected to a handpiece. This procedure does not necessitate the use of scalpels even in the presence of dense breasts or hard and fibrous specimens, thanks to the needle’s high capacity of penetration and manageability, which also allows good system control during the procedure and a 360° rotation within the biopsy area typical of the VABB.

Samples obtained using Elite, more than three times the weight and volume compared to CB (approx. 60 vs. 12 g), are collected automatically and extracted, avoiding any external contact, into a small radio-transparent container fixed to the base of the handpiece (Figure 3 and Figure 4). The samples are fixed in formalin and sent to the Department of Pathological Anatomy. At the end of the procedure, in order to relocate the site, a small non-magnetic clip may be inserted, which is visible in mammography and ultrasound for up to a maximum of 6 months, in cases of lesions that could mutate over time, such as complex cysts or tumors needing neoadjuvant therapy (Figure 5).

### 2.3. Statistical Analysis

The diagnostic performances of the Elite VABB procedure were compared to those obtained with the CB. In particular, the diagnostic performances were estimated compared to the results of each procedure with the conclusive histological result in terms of accuracy (correct sampling), sensitivity, specificity, and diagnostic underestimation rate. Specifically, accuracy was the number of correct results out of the total number of samples, sensitivity was the percentages of benign cases (results B1, B2, B3), and specificity was malignant cases (results B4, B5) classified correctly. The diagnostic underestimation rate was estimated as the number of underestimated cases of the true nature and extent of a biopsied lesion in relation to the post-operative histological results (gold standard). Moreover, the radio-pathological concordance using different needles was assessed, compared to the conclusive histology and operator-related variability. Finally, the results of the VABB Elite procedure were compared to those of the CB and FNAC procedures carried out on the same patients in order to statistically assess the concordance of the procedure. The estimated rate of inadequacy was measured as the ratio of the class 1 results in relationship with the total procedures performed and the rate of inconclusive results as the ratio of the class 1 and class 3 results in relationship with the total number of procedures performed.

## 3. Results

The Elite system allowed the typification of 200 ultrasound biopsies of dimensions ranging between 3 and 55 mm including 8 complex cysts, 35 areas with microcalcifications, 32 distorted areas or areas with echo structural alterations, and 125 lumps.

Table 1 summarizes the characteristics of the tumors analyzed using the Elite VABB and CB systems. All the Elite VABB biopsies concerned lesions classified by ultrasound as doubtful, suspected, or suggested malignancy (BI-RADS classes 3, 4, or 5, respectively).

The reasons for the use of the Elite VABB system were associated with difficult CB breast biopsies (34.50%), the need for biopsy specimens of larger dimensions in non-mass lesions with undefined margins (27.50%), first cyto-histological result indeterminate (19.50% of cases), excisional biopsy (15.00%), or radio-pathological discrepancy (3.50% of cases). The biopsies with an initial radio-pathological discrepancy (U4/5 and micro-histological result B1/B2) were reassessed by a second operator and finally repeated.

Following the Elite biopsy, 100 patients underwent surgery, that is, 26 out of the 43 patients with a B3 result, 2 patients with a B4 result, and 70 out of the 71 patients with B5 lesions (1 patient with a B5 result did not undergo surgery due to critical clinical conditions and later died). One patient with a B1 result and another with a B2 result, respectively, underwent surgery owing to suspected radiological malignancy and the patient’s personal choice following inconsistencies with previous class 3 cytological micro-histological biopsies. Among the patients at follow-up after the first CB biopsy, 7 with a B3 result presented a variation in the radiological picture between 6 and 12 months after the biopsy and underwent surgery, highlighting 5 cases of diagnostic underestimation of B5 (1 invasive ductal carcinoma and 4 DCIS) and 2 cases of atypical ductal hyperplasia (B3). No modifications in the clinical picture were registered in the remaining cases at follow-up in both the CB and the VABB patients.

Of the 1314 biopsies performed using the CB procedure, 648 of the 650 B5 cases, the 16 B4 cases, and 82 of the 121 B3 cases, for a total of 746 cases, underwent surgery: 2 women with a B5 result died prior to surgery.

Table 2 shows the details of the conclusive histological results of the 100 Elite VABB patients and of the 746 CB patients who then underwent surgery. The Elite VABB procedure shows an elevated diagnostic performance with an accuracy of correct sampling of 94.00% (2 with B2, 20 with B3, 2 with B4, and 70 with B5, all correctly diagnosed), with a sensitivity of 92.30% and a specificity of 100%. With reference to the CB procedure, the estimate of correct sampling was equal to 90.58% with a sensitivity of 96.37%, a specificity of 100%, and a diagnostic performance comparable to that of the Elite VABB procedure.

The diagnostic underestimation for the Elite VABB procedure was 11.00% and concerned only 5 cases of B3 (4 DCIS and a K tubular) that turned out to be B5 in conclusive histology and 6 cases passed at biopsy from DCIS to invasive ductal carcinoma (IDC). Such underestimation is most likely connected to the difficult visualization and the scarce margins of a particular type of lesion under ultrasound. The diagnostic underestimation for the CB procedure was equal to 25.26% and concerned 153 cases in class B5 (DCIS in CB resulting IDC and invasive lobular carcinoma (ILC) at conclusive histology), 16 cases in class B4 (6 IDC and 10 DCIS at conclusive histology) and 24 cases in class B3 (18 DCIS and 6 IDC).

The diagnostic performance in terms of correct sampling and underestimation of the Elite VABB procedure is significantly comparable to that observed using the CB procedure (*p*-value test *T* > 0.01).

Figure 6 and Figure 7 summarize the experimental results regarding the radio-pathological concurrence of the respective biopsies performed with CB and the Elite VABB procedures, differentiated also by operator (A and B) and needle gauge used during the CB procedure (14G and 16G). According to BI-RADS criteria, we considered the radio-pathological classes 1,2,3 indicative of benignity and the radio-pathological classes 4,5 indicative of malignancy, therefore we verified the concordance or the discordance between the pre-biopsy radiological evaluation and the histological outcome. The radiological class 3 is, indeed, a definite category “probably benign” with positive predictive value (VPP) < 5%, while class 4 includes lesions with differing levels of suspected malignancy with a generally elevated VVP. The same thing can be said for classes B3 and B4 at biopsy in relation to conclusive histology, even with the necessary limitations connected to underestimation regarding extended lesions, particular types, and insufficient specimens [14,15,16,33,34].

In our samplings, a result was considered to be concordant when both the radiological and histological evaluation (post-biopsy) fell into the same binary class (benign/benign or malignant/malignant); while in the other cases (benign/malignant) it was considered discordant. The study shows substantial overlapping (*p*-value test *T* > 0.01) in the total radio-pathological concurrence of the results between Elite VABB and CB (77.00% vs. 87.44%), between the operators in both the CB (87.13% vs. 87.66%) and the Elite VABB procedure (79.34% vs. 75.00%) and also in the use of different 14G and 16G needles in the CB procedure (87.74% vs. 86.26%). This highlights a good inter–intra-observer concurrence in this type of analysis between the different types of needles used.

Table 3 shows the results of Elite VABB procedure as compared to the post-operative conclusive histology results correlated by the operator who carried out the procedure. Operator B showed a lower correct sampling accuracy (85.71% vs. 100%) and also a higher rate of inconclusive results (class 1 or 3) than operator A (3.17% vs. 0%). The same was also true for the CB procedure, operator B again showed a lower correct sampling accuracy compared to operator A (85.25% vs. 92.62%) and a higher rate of inconclusive results (class 3) than operator A (7.76% vs. 4.90%).

Of the 196 patients who underwent a micro-histological biopsy with the Elite VABB procedure, 31 had previously undergone a CB biopsy using semi-automatic 14–16G core needles, 18 at first-level cytological assessment by means of FNAC, and 3 underwent both techniques. Table 4 summarizes the results of the CB and FNAC procedures compared to those obtained in the successive Elite VABB procedure. The number of cases that modified the initial cytological class were 19/21 (90.47%), whereas the cases that did not change the initial micro-histological class were 23/34 (67.64%). In particular, in 19 cases, the doubtful cyto-histological biopsy (C3/B3) changed to benign (B2) which, in concurrence with the imaging, avoided surgery; in the 6 cases with an inadequate cyto-histological biopsy (C1/B1), the diagnostic result was reconsidered using Elite VABB, which correctly identified the histological class, that is to say 4 cases of B2 and 2 cases of B5. Among the complications of the procedures were 2 cases of prolonged bleeding and 10 of pain experienced in the Elite VABB procedure and 8 and 10 cases in the 14G CB, respectively.

The CB procedure in the sample analyzed highlighted an inadequacy rate of 1.98% and a rate of inconclusive results of 11.19% (Figure 8). The Elite VABB system highlighted a contained inadequacy rate of 1.50%, in line with published results (1.22% B1 on 10G Encore and 0% on 11G Mammotome, 1.40% B1 on ultrasound-guided Elite VABB) [14,15,35,36], and a rate of inconclusive results of 23.00%, both significantly comparable to those of the Elite procedure (*p*-value *T* test >0.01).

## 4. Discussion

Our Elite VABB procedures were of easy technical execution thanks to good handling, the lightness of the handpiece, the intuitiveness of the controls, and the excellent penetrating capacity of the needle even in the tissues of greater density, better than the CB. Incomplete procedures were not reported due to technical problems which, even so, did not obstruct the procedure. Moreover, there was a substantial overlap in the cases of bleeding between CB and Elite (respectively, 0.60% and 1.00%) while Elite showed a relatively higher number of patients experiencing pain during the procedure (0.76% vs. 5.00%) in line with reported literature [37]. The Elite VABB system, thanks to its automatic suction system (TruVac), avoids contact with the biopsied samples and bleeding tissue permitting the direct passage into the posterior collection tray, which can be used for transport to the Pathological Anatomy Department avoiding any additional steps.

The Elite VABB procedure when compared to the CB in samples examined by us highlighted a lower inadequacy rate (1.50% vs. 1.98%) consistent with results already published in literature (1.22% B1 for 10G Encore and 0% for11G Mammotome, 1.40% B1 for ultrasound-guided Elite VABB) [14,15].

The comparison between the results in the inconclusive class, which are relatively higher in the Elite VABB (23.00% vs. 11.19%) than in the CB, is partly explained by the particular types of lesions selected in the former. It does not have any impact on the statistical assessment where the two systems, given the different number of samples, appear comparable (*p*-value *T* test > 0.01).

The microcalcification clusters, which are indicated in the VABB biopsy, if visible at ultrasound, are also easily and immediately sampled with Elite VABB, thus reducing the percentage of the stereotaxic VABBs and promoting a saving in terms of cost, time, and personnel (due to the absence of a radiological technician). This acknowledgement could increase further in the future in the event of development in automatic systems of detection and characterization of such specimens even in ultrasound as present in mammography [38,39,40,41,42].

Moreover, our study underlined an overall radio-pathological discordance of 12.55% in the CB cases and 23.00% in the Elite VABB cases: this does not permit us to create a safe diagnostic pathway in the presence of a significant percentage of indeterminate first-line biopsy results.

In literature, an instrument to assess the effectiveness and the accuracy of a VABB system is that of the underestimation of lesions considered at risk in the B3 classes namely atypical ductal hyperplasia (ADH) [14].

Bennet [17] reported a recovery of 23% of malignant lesions by performing the VABB procedure after a radio-pathological discrepancy had been found in a CB sampling of lesions, thereby recommending its use on extensive areas of microcalcifications and on lesions smaller than 5 mm. In accordance with this observation, we decided to examine all lesions with dimensions inferior to 5 mm with Elite VABB.

Mayer [43] showed a significantly higher percentage of malignant tumors in the B3 lesions with atypia (B3b) subjected to surgical resection compared to B3 lesions without atypia, namely B3a (24.00% vs. 5.80% including the papillary lesions).

The abovementioned considerations would suggest excisional biopic surgery in several cases of B3 lesions after CB, especially in the presence of atypia. In the remaining cases of B3, the absence of radiological suspicion would be insufficient to exclude neoplastic lesions in a significant percentage of cases. Therefore, the most suitable choice might actually be the use of a second-line VABB procedure, after multidisciplinary discussion [29,43,44,45].

A single preliminary study [46] on a cable-free VABB system reported potential disadvantages in the diagnosis of non-nodular lesions (non-mass). Successive studies in literature on comparable systems to the present study [25,37,46,47,48,49,50,51] have not shown any significant differences in terms of sensitivity, specificity, and diagnostic accuracy compared to the traditional VABB using an 8–11G needle but only a greater underestimation and a larger number of FNs [25], probably correlated to the gauge of the smaller needle.

As far as we know, this represents the first study of the direct correlation between Elite 13G VABB and 14–16G CB on breast lesions using ultrasound with the same operators.

In only one other study [37], a comparative assessment was carried out between Elite 13G VABB and 14G CB but this case was on axillary lymph nodes in patients with pre-operative breast cancer with results for sensitivity, inadequate numbers, and cases of hematoma that practically overlapped in the two procedures. Moreover, the Elite VABB procedure showed a moderately lower patient acceptability compared to the CB procedure (85% vs. 97%) mainly correlated to a greater level of post-bioptic pain, coherent with the observations in our study.

This type of Elite VABB also appears to highlight a lower underestimation as compared to the CB procedure, particularly in the lobular forms in situ (lobular cancer in situ, LCIS) and higher in the non-nodular lesions (non-mass) [52].

Moreover, the tested systems [47] satisfy the criteria of accuracy suggested by the European guidelines and according to some estimates, this type of ultrasound biopsy does not incur higher costs for the diagnosis of cancer compared to those of CB [51].

The choice of surgery for the B3 classes was connected to the type of lesion found, the presence of atypia in the first biopsy, the presence of the patient’s familial or personal risk factors, and the persistent radio-pathological discrepancies. The patients who did not have surgery underwent an instrumental follow-up at 6 months by means of ultrasound, a clinical-mammographic-ultrasound follow-up at 12 months and subsequently annually. The higher percentage of B3 lesions found in the Elite 13G VABB (21.50%) despite the larger gauge compared to the 14–16G CB results (9.21%, consistent with most literature), could be associated with the particular type of lesion selected for the biopsy with the first system, a higher percentage in Elite VABB compared to a CB. From this point of view, the image fusion system of Magnetic Resonance Imaging (MRI) and Ultrasound (US) imaging could be useful for a better identification and definition of the margins of some of these lesions [52]. However, the good quality of the samples was confirmed by the low number of inadequate biopsies (B1) and of B4 lesions in the samples taken with the Elite VABB.

As regards the second-level samples, the Elite VABB system made it possible for a high percentage of largely inconclusive samples (90.47% cytological and 67.64% CB) to change histological class with a final result reflecting a better diagnostic definition, which would suggest the important role of this technique in doubtful or complex diagnoses.

In general terms, in our experience, the Elite 13G VABB system has proven to be high performing with a high estimation of correct sampling (94.00% vs. 90.58% CB) and a contained underestimation (11.00% vs. 25.26% CB). Considering the different sample sizes, the study, even so, shows substantial statistical overlapping of the techniques in total concordance both radio-pathologically and between operators: this would highlight a good inter/intra-observer concordance in this type of analysis even as regards the different types of needle used.

One of the limitations of the present study comprises the different number of the sampled cohorts among the techniques used, both in comparison between the CB and the Elite VABB and in relation to the CB, considering the different types of needle used. Furthermore, it is also necessary to assess the diagnostic performance among each of the different Elite VABB needle gauges on the market (10G and 13G).

## 5. Conclusions

The bioptic procedure using the Elite 13G VABB system has proven to be a simple, quick, reliable procedure well-tolerated by patients with no significant complications. The two biopsy techniques showed similar reliability on many samples without significant changes in age classes. Based on what is reported in literature on VABB systems and based on our personal experience, we recommended the use of Elite 13G VABB as a first-line biopsy on microcalcifications, distortions, papillary lesions, or eco-structural alterations with undefined contours: this is due to the inherent possibilities of this method of larger sampling, in multiple positions and directions.

The use of increasingly high-performance ultrasound with high-frequency linear probes, which allows the imaging of a greater number of distortions and microcalcifications, could bring about, in the near future, an alternative bioptic approach compared to the stereotaxic one with a saving in cost and personnel.

The change in the histological class of a great number of repeated biopsies would suggest an important role of Elite VABB 13G as second-line biopsy in cases of doubt or when faced with indeterminate cyto-histological results.

## Figures and Tables

**Figure 1 diagnostics-10-00291-f001:**
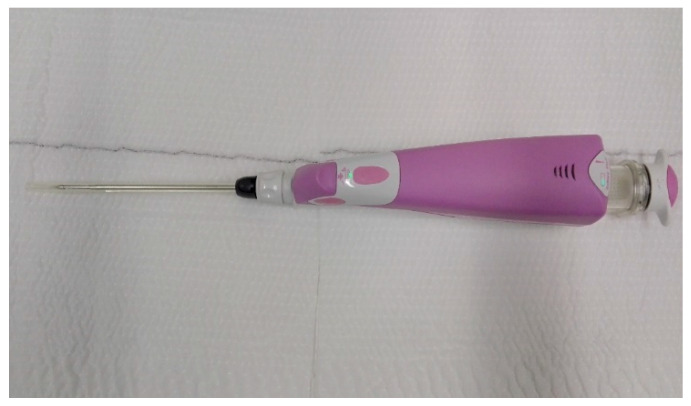
Vacuum-assisted breast biopsy (VABB) Elite ultrasound-guided biopsy handpiece, manageable and slightly larger than a core biopsy (CB) handpiece.

**Figure 2 diagnostics-10-00291-f002:**
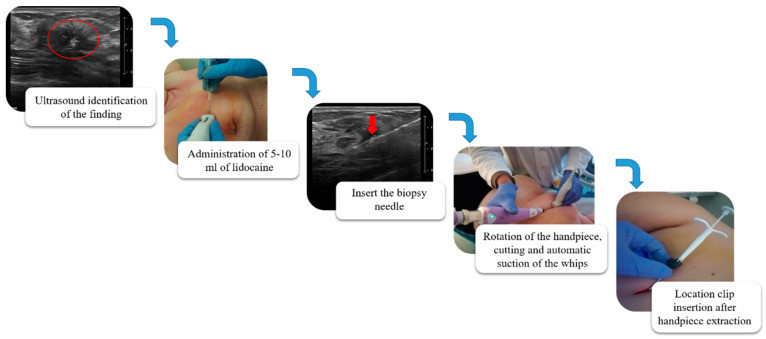
Workflow of the VABB Elite procedure. The biopsy finding is initially identified on the ultrasound; 5–10 mL of lidocaine s.c. is then administered (perilesional subcutaneous); the biopsy needle is visualized by ultrasound (red arrow in the figure) until the sampling site is reached; the procedure is started with a 360° rotation at the pick-up, cutting and automatic suction of the whips in the collection container located at the base of the handpiece; at the end of the procedure it is possible to extract the needle and the handpiece, leaving in place a cannula for the insertion of a localization clip visible in ultrasound and mammography up to 6 months after the procedure.

**Figure 3 diagnostics-10-00291-f003:**
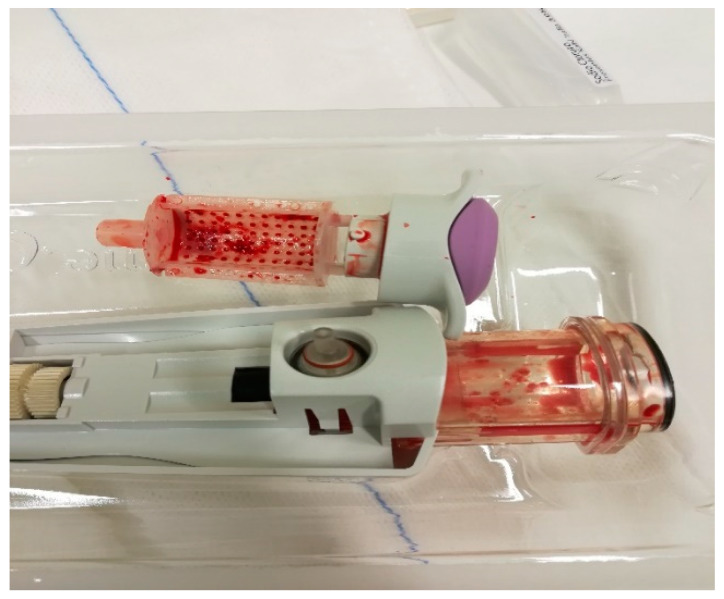
Samples collected automatically in a small easily extractable radio-transparent container using the CB system.

**Figure 4 diagnostics-10-00291-f004:**
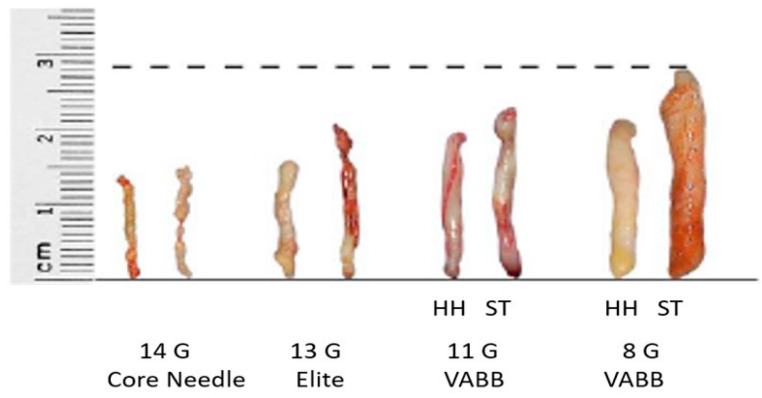
Reference scheme for average weight and size of whips in the main currently used ultrasound (HH) and stereotaxic (ST) bioptic techniques: the differences in volume and weight among the 14G CB (12 g), 13G VABB Elite (60 g) 11G VABB Mammotome (≈100 g) and 8G VABB Mammotome (≈300 g) samples are shown.

**Figure 5 diagnostics-10-00291-f005:**
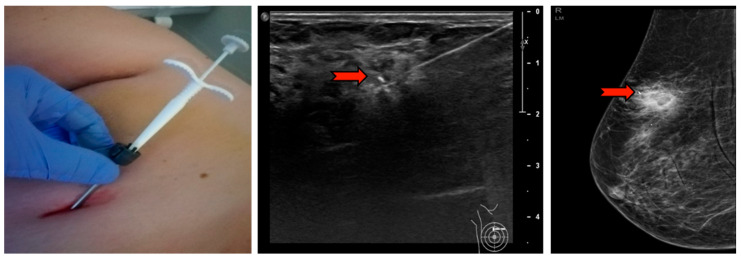
The needle and handpiece extraction and subsequent insertion of the localizing clip in the cannula left in the biopsy site. The red arrow indicates the clip in mammography and ultrasound that will be visible in follow-ups for up to 6 months.

**Figure 6 diagnostics-10-00291-f006:**
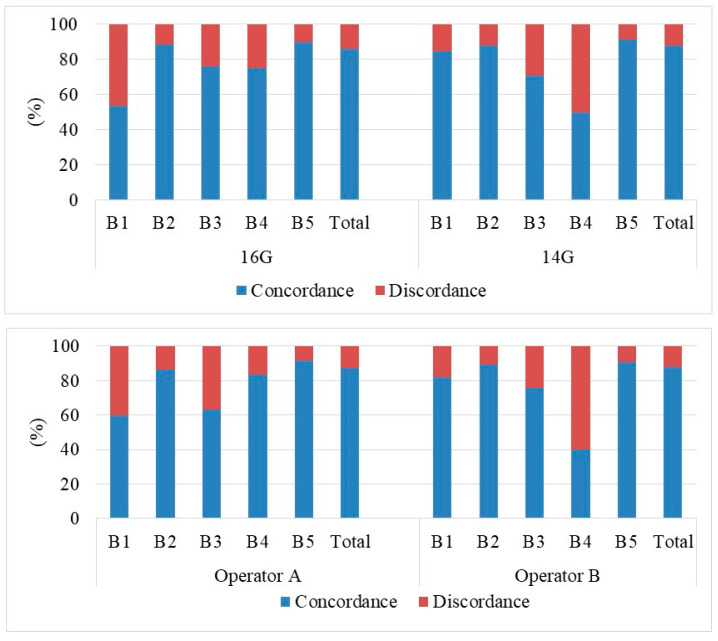
Assessment of the radio-pathological concurrence compared to the post-operative conclusive histology results of patients who underwent the CB procedure.

**Figure 7 diagnostics-10-00291-f007:**
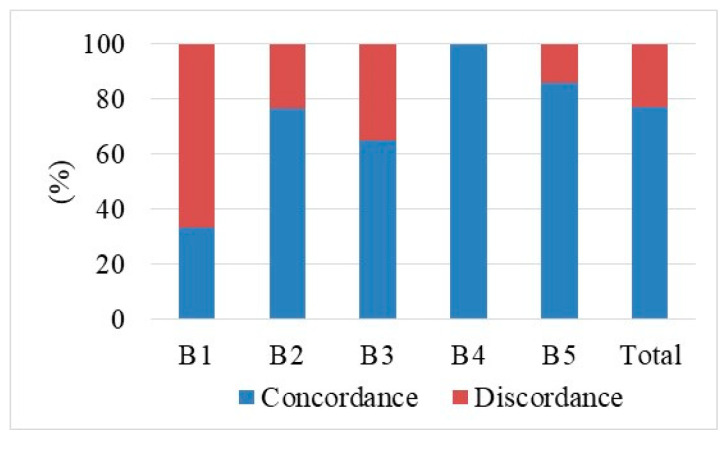
Assessment of the radio-pathological concurrence compared to the post-operative conclusive histology results of patients who underwent the Elite VABB procedure.

**Figure 8 diagnostics-10-00291-f008:**
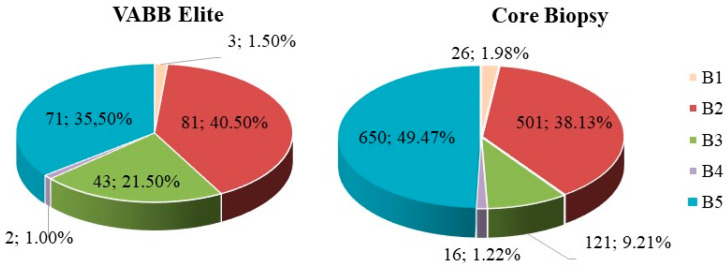
Percentage distribution (and absolute values) of the micro-histological biopsy results carried out by means of the two procedures Elite VABB and CB using 14–16G core needle.

**Table 1 diagnostics-10-00291-t001:** Characteristics of the tumors analyzed using the Elite VABB and CB system.

Characteristic	No. of Lesions VABB Elite	No. of Lesions CB
Histological subtype		
Invasive ductal carcinoma (IDC)	46	454
Invasive lobular carcinoma (ILC)	11	93
Ductal carcinoma in situ (DCIS)	7	72
Other cancer	7	31
Histological grade (Nottingham scale)		
High (G1)	18	101
Intermediate (G2)	39	405
Low (G3)	14	144
Tumor size		
<10 mm	32	205
10–19 mm	22	256
20–29 mm	10	137
≥30 mm	7	52

**Table 2 diagnostics-10-00291-t002:** Results of the Elite VABB (100 patients) and CB (746 patients) procedures as compared to the post-operative conclusive histological results.

		VABB Elite					
		B1	B2	B3	B4	B5	Total
Conclusive Histology	B2		2 (100%)				2 (2.00%)
	B3			20 (80.00%)			20 (20.00%)
	B5	1 (100%)		5 (20.00%)	2 (100%)	70 (100%)	78 (78.00%)
	Total	1 (100%)	2 (100%)	25 (100%)	2 (100%)	70 (100%)	100 (100%)
**CB**							
		**B1**	**B2**	**B3**	**B4**	**B5**	**Total**
Conclusive Histology	B2			13 (15.85%)			13 (1.76%)
	B3			44 (53.66%)			44 (5.90%)
	B5			25 (30.49%)	16 (100%)	648 (100%)	689 (92.44%)
	Total			82 (100%)	16 (100%)	648 (100%)	746 (100%)

**Table 3 diagnostics-10-00291-t003:** Details of the results of the Elite VABB (100 patients) and CB (746 patients) procedures compared to the post-operative conclusive histology results correlated by the operator who carried out the procedure.

		VABB Elite											
		Operator A						Operator B					
		B1	B2	B3	B4	B5	Total	B1	B2	B3	B4	B5	Total
Conclusive histology	B2		1 (100%)				1 (2.70%)			1 (5.56%)			1 (1.57%)
	B3			8 (100%)			8 (21.63%)			12 (66.67%)			12 (19.06%)
	B5					28 (100%)	28 (75.67%)	1 (100%)		5 (27.77%)	2 (100%)	42 (100%)	50 (79.37%)
	Total		1 (100%)	8 (100%)		28 (100%)	37 (100%)	1 (100%)		18 (100%)	2 (100%)	42(100%)	63 (100%)
**CB**													
	**Operator A**						**Operator B**						
		**B1**	**B2**	**B3**	**B4**	**B5**	**Total**	**B1**	**B2**	**B3**	**B4**	**B5**	**Total**
Conclusive histology	B2			5 (20.83%)			5 (1.47%)			8 (13.80%)			8 (1.97%)
	B3			12 (50.0%)			12 (3.54%)			32 (55.17%)			32 (7.85%)
	B5			7 (29.17%)	6 (100%)	309 (100%)	322 (95.00%)			18 (31.03%)	10 (100%)	339 (100%)	367 (90.18%)
	Total			24 (100%)	6 (100%)	309 (100%)	339 (100%)			58 (100%)	10 (100%)	339 (100%)	407 (100%)

**Table 4 diagnostics-10-00291-t004:** Results of the Elite VABB procedure compared to the CB (34 patients) and fine-needle aspiration cytology (FNAC) (21 patients) procedures of patients who underwent both procedures.

		CB					
		B1	B2	B3	B4	B5	Total
VABB Elite	B1			1 (4.55%)			1 (2.94%)
	B2	3 (50.00%)	2 (40.00%)	11 (50.00%)			16 (47.06%)
	B3		2 (40.00%)	9 (40.91%)			11 (32.35%)
	B4						
	B5	3 (50.00%)	1 (20.00%)	1 (4.55%)	1 (100%)		6 (17.65%)
	Total	6 (100%)	5 (100%)	22 (100%)	1 (100%)		34 (100%)
		**FNAC**					
		**C1**	**C2**	**C3**	**C4**	**C5**	**Total**
VABB Elite	B2	1 (33.33%)		8 (44.44%)			9 (42.86%)
	B3			1 (5.56%)			1 (4.76%)
	B4			2 (11.11%)			2 (9.52%)
	B5	2 (66.77%)		7 (38.89%)			9 (42.86%)
	Total	3 (100%)		18 (100%)			21 (100%)

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
