# Peer review of "Elite VABB 13G: A New Ultrasound-Guided Wireless Biopsy System for Breast Lesions. Technical Characteristics and Comparison with Respect to Traditional Core-Biopsy 14–16G Systems"

_diagnostics, 2020, doi:10.3390/diagnostics10050291_

Round 1
Reviewer 1 Report
I suggest that the results in Table 2,3 and 4 should be presented also in % not only in numbers.
Please describe more clearly , in the title of each Table, regarding the total number for each category (CB, FNAC, VABB)
It is difficult to have any conclusion of the 2 different biopsies techniques were offered in differed patients, considering age, disease type....
Please do explain more clearly the concordance with the pathology report, Sensitivity, Specificity, Accuracy for each method should be discussed. The results of FNAC, offered for less than 50 cases out of 1400 cases in completely non significant....
Author Response
Response to Reviewer 1 Comments
Point 1. I suggest that the results in Table 2,3 and 4 should be presented also in % not only in numbers.
Response 1. As suggested by the reviewer, we also calculated the percentage values in Table 2,3 and 4.
Point 2. Please describe more clearly , in the title of each Table, regarding the total number for each category (CB, FNAC, VABB)
Response 2. As suggested by the reviewer, we also indicated the total number of samples for each procedures.
Point 3. It is difficult to have any conclusion of the 2 different biopsies techniques were offered in differed patients, considering age, disease type....
Response 3. We thank the reviewer whose comments helped us to clarify the same points of the manuscript. We have clarified this specific aspect in the Conclusions section.
Point 4. Please do explain more clearly the concordance with the pathology report, Sensitivity, Specificity, Accuracy for each method should be discussed. The results of FNAC, offered for less than 50 cases out of 1400 cases in completely non significant....
Response 4. In the Statistical analysis subparagraph, we have specified how accuracy, sensitivity and specificity have been estimated. With reagrd to the results of FNAC, Table 4 summarizes the results of the cases that performed Elite VABB 13 G as the second biopsy line after CB (n = 34) or FNAC (n = 21). The numbers of this patient cohort are relatively low compared to the overall number of Elite VABB 13 G performed (n = 200) and CB 14-16G procedures (n = 1314) and we agree with the reviewer that these numbers should be implemented in the future. However, we consider this a first preliminary evaluation on a relatively new technique that can provide us with some initial hypotheses on the use of Elite VABB 13 G which has shown itself in line with what has already been recommended in the literature on ultrasound and stereotaxic VABB systems at a higher caliber.
Reviewer 2 Report
The authors present an Elite 13G VABB system that provides noticeable advancement over state-of-the-art methods. The results are well quantified and the results are satisfactory. However, the paper needs to be improved to ensure publication. The primary comments are as follows:
1) Instead of just showing images of procedures, provide sketches/ workflow of the process together with the images of the procedural steps to make the methods more self-explanatory to a larger audience.
2) The processes are ultrasound-assisted, please provide ultrasound tracking images wherever possible for reference (incl. Fig 2).
3) Fig 4 caption says it compared volume and weight -- it does none. Provide scale bars to compare size, add details of weight and volume comparison in separate sub-figure as graphs/ volume distribution visualizations.
4) Fig 5. Show the followup mammographic and US images as mentioned.
5) Fig 6 and 7 : What is Concordia and Discordi? Please translate if needed and provide details in the text.
6) The literature review is only in the conclusion section. Please move the same to the introductory sections. In the Discussions section please only discuss results and only compare with the literature values.
7) Please describe the novelty of the system in more detail in the introduction and quality of them in the results and discussion.
The writing is very unorthodox and difficult to follow as authors discuss the literature only in discussion/ conclusion -- so the format and writeup need major revision. The attention to detail for images in minimal which is disappointing. The overall readability and information conveyed by the figure (and captions) needs to be improved significantly.
Author Response
Response to Reviewer 2 Comments
The authors present an Elite 13G VABB system that provides noticeable advancement over state-of-the-art methods. The results are well quantified and the results are satisfactory. However, the paper needs to be improved to ensure publication. The primary comments are as follows:
Point 1. Instead of just showing images of procedures, provide sketches/ workflow of the process together with the images of the procedural steps to make the methods more self-explanatory to a larger audience.
Response 1. As suggested, we have introduced a workflow of the process.
Point 2. The processes are ultrasound-assisted, please provide ultrasound tracking images wherever possible for reference (incl. Fig 2).
Response 2. In Figure 2, we have introduced a workflow of the process as requested in the previous point where ultrasound tracking images are also provided. We are not sure if we correctly understood the auditor's request. However, we are available to review it on the indication of the same.
Point 3. Fig 4 caption says it compared volume and weight -- it does none. Provide scale bars to compare size, add details of weight and volume comparison in separate sub-figure as graphs/ volume distribution visualizations.
Response 3. We have replaced the image of figure 4 with a more explanatory one by providing a bar to compare the dimensions and indicating in the caption the weight of the samples for each procedure.
Point 4. Fig 5. Show the followup mammographic and US images as mentioned.
Response 4. We have replaced the image of figure 4 with a more explanatory one by adding mammographic and ultrasound images.
Point 5. Fig 6 and 7: What is Concordia and Discordi? Please translate if needed and provide details in the text.
Response 5. Tables 6 and 7 represent the radio pathological concordance / discrepancy results. We have considered the radio-pathological classes 1,2,3 indicative for benignity and the radio-pathological classes 4,5 indicative for malignancy, therefore we have verified the concordance or the discordance between the pre-biopsy radiological evaluation and the histological outcome. For example, if the radiologist evaluated a BIRADS 3 finding and the histological result was B2, the judgment was considered concordant, conversely if the histological result was B5, the judgment was discordant. We evaluated this by single procedure (CB fig. 6 - Elite fig. 7) and, within the same procedure, by operator.
In Results section after table 2, we had reported the explanation of this. However, according to the suggests of the reviewern, in order to make the concepts clearer, we have provide some details in the text.
Point 6. The literature review is only in the conclusion section. Please move the same to the introductory sections. In the Discussions section please only discuss results and only compare with the literature values.
Response 6. We thank the reviewer whose suggestions helped us to clarify the same points of the manuscript. We revised the Introduction and discussions sections following your suggestions
Point 7. Please describe the novelty of the system in more detail in the introduction and quality of them in the results and discussion.
Response 7. The VABB systems could represent a good compromise between performance and manageability thanks to a new generation of 13 G VABB needles has been introduced that can be used by ultrasound with a slightly bigger handpiece than a CB needle. The Elite VABB procedure when compared to the CB in samples examined by us highlighted a lower inadequacy rate. In Introduction and Discussion sections, we have explained in more detail the novelty and results quality as suggested by reviewer.
Point 8. The writing is very unorthodox and difficult to follow as authors discuss the literature only in discussion/ conclusion -- so the format and writeup need major revision. The attention to detail for images in minimal which is disappointing. The overall readability and information conveyed by the figure (and captions) needs to be improved significantly.
Response 8. Thanks to your suggestion, we reviewed the introductory part and the discussions by reporting the main literature exposure in the first and resetting the second. Moreover, we have also carried out a general review of the manuscrip, included figures, tables and their captions. A native English speaker has reviewed the manuscript.
Point 9. I suggest that the results in Table 2, 3 and 4 should be presented also in % not only in numbers.
Response 9. As suggested by the reviewer, we also calculated the percentage values in Table 2,3 and 4.
Reviewer 3 Report
The topic of the paper is interesting and meets the aims and scope of the journal but it has some problems
The topic of the paper is interesting and meets the aims and scope of the journal but it has some problems
However, I have some suggestions for the paper improvement, as follows:
Introduction:
I think that there is a need for reviewing similar researches and should mention the results. With this way the researchers should demonstrate the need of this study. Also, the authors should write research hypotheses.
The topic of the paper is interesting and meets the aims and scope of the journal but it has some problems
in the discussion the better presentation is you write first your results and comment them with other study results
Author Response
Response to Reviewer 3 Comments
The topic of the paper is interesting and meets the aims and scope of the journal but it has some problems. However, I have some suggestions for the paper improvement, as follows:
Point 1. Introduction: I think that there is a need for reviewing similar researches and should mention the results. With this way the researchers should demonstrate the need of this study. Also, the authors should write research hypotheses.
Response 1. We thank the reviewer for his suggestions. We have included most of the state of the art in the Introduction section with the aim of better answering to his comments.
Point 2. In the discussion the better presentation is you write first your results and comment them with other study results
Response 2. We welcome the advice proposed by the auditor. According to these indications, the Discussion section was structured.
Round 2
Reviewer 1 Report
I do not have further comments.
Reviewer 2 Report
The authors have responded suitable to the reviewers and the article may be accepted. The authors are requested to thoroughly proofread the article to iron out minor textual and grammatical errors.